# Effectiveness of a Humor-Based Training for Reducing Employees’ Distress

**DOI:** 10.3390/ijerph182111177

**Published:** 2021-10-24

**Authors:** Jose M. León-Pérez, Francisco J. Cantero-Sánchez, Ángela Fernández-Canseco, José M. León-Rubio

**Affiliations:** Department of Social Psychology, Universidad de Sevilla, 41018 Sevilla, Spain; leonperez@us.es (J.M.L.-P.); fcantero@us.es (F.J.C.-S.); fernandezcanseco.maria@gmail.com (Á.F.-C.)

**Keywords:** sense of humor, coping strategies, cheerfulness, seriousness, positive psychology, psychological well-being, social learning theory

## Abstract

An increasing number of studies have demonstrated that humor can impact interpersonal relationships in organizations and employee well-being. However, there is little evidence coming from intervention studies in organizational settings. In response, we developed a training following the principles of positive psychology that aims at improving employees’ adaptive use of humor as a successful mechanism to deal with stress. In this study, we assess the effectiveness of such training and its impact on employee well-being. Results from this one-group intervention study in an emergency ambulance service (N = 58) revealed that the participants reported higher levels of cheerfulness (*Z* = −3.93; *p* < 0.001) and lower levels of seriousness (*Z* = −3.32; *p* < 0.001) after being exposed to the training. Indeed, the participants reported lower scores on psychological distress after the training (*Z* = −3.35; *p* < 0.001). The effect size of the training was medium (*r* = 0.31 to 0.36), suggesting that interventions to improve adaptive humor at work can be a useful resource to deal with workplace stress and foster employee well-being. These results may have interesting implications for designing and implementing positive interventions as well as for developing healthy organizations.

## 1. Introduction

Organizations provide one of the main sources of stress that people experience in their lives due to the amount of time exposed to job pressures such as work overload and negative social interactions [1]. Therefore, workplace stress is becoming a significant public health problem that is associated with numerous negative consequences at different levels, from individuals’ health-related issues such as higher risk of heart disease and suffering type-2 diabetes [2,3,4,5], to increased social security costs [6]. Among occupations, workers in the healthcare sector seem at risk as they show one of the highest levels of workplace stress compared to all other workers [7]; a picture that seems to be even worse in current times due to the COVID-19 crisis [8].

Therefore, stress management seems crucial for both improving employees’ health and well-being and successfully perform job tasks, providing a high-quality service. In this regard, several meta-analyses have shown the effectiveness of workplace stress management interventions [9,10]. For example, Richardson and colleagues [9] analyzed the effectiveness of 55 interventions in occupational settings aimed at decreasing workers’ stress levels (N = 2847). Their results revealed a significant medium to large effect of the interventions. Furthermore, they found a larger effect for interventions with a cognitive–behavioral approach compared to other methods for managing workplace stress that included several components (i.e., multimodal interventions) or were based on meditation, relaxation, and deep-breathing techniques. In contrast, Kröll and colleagues [10], after analyzing 21 primary studies, reported larger effects for relaxation interventions compared to cognitive–behavioral and multimodal interventions when psychological health outcomes were considered. In a similar vein, a recent meta-analysis on randomized control trials has shown that mindfulness training programs have beneficial consequences for employees’ psychological health and well-being, effects that last several months after the training [11].

In the last years, beyond more traditional stress management interventions, some alternative interventions rooted in Positive Psychology have been proposed (for meta-analyses, see [12,13,14]). Indeed, there is growing evidence suggesting that a positive use of humor can be considered an effective coping resource to get rid of stress, and humor has been associated with psychological well-being at work (for a meta-analysis, see [15]). Humor is a complex phenomenon that comprises several aspects, from the insight to perceive and understand the humorous, laughable, or ridiculous side of things to the mental capacity to create or establish unusual relationships that surprise and make others laugh. Humor can also refer to a state of mind that predisposes one to be cheerful and optimistic and generate positive feelings and emotions in others. In addition, from a communication perspective, humor is considered as a pleasant mode of communication that elicits humorous reactions and expressions such as laughter.

Researchers have followed a multidimensional conception of humor [16,17,18], when humor has been addressed as a moderating variable in the relationship between stress and mood and well-being [19,20], which includes dimensions such as the ability to create or generate humor, the capacity to face life’s problems and difficulties with optimism and cheerfulness, the ability to appreciate and enjoy the humor generated by others, and the attitude or disposition to act and communicate through humor. This is the perspective we assumed in this study to examine the effectiveness of a training program aimed at promoting an adaptive humor at work as a mechanism to deal with daily workplace stress in emergency healthcare professionals. Indeed, although humor-based interventions have been proposed and proved to achieve good outcomes [21,22,23,24,25,26,27,28], few studies have addressed humor-based stress reduction interventions in organizational settings.

### Humor as a Stress Coping Mechanism

The assumption that humor can be an effective resource for coping with stress is based on three core elements. First, from the perspective of a cognitive approach, transactional theories of stress consider that stress is elicited by one’s appraisal of a situation. In other words, stress reactions and consequences depend on the extent that a stressor is perceived as threatening something that is considered valuable, as an opportunity for growth, or as a work challenge because the person has the necessary skills and resources to deal with the situation [29]. Thus, considering that people experience less distress when situations are perceived as a challenge rather than as a threat, humor has a stress-buffering effect because it helps people reappraise stressful events as challenges [29,30,31,32,33,34,35]. For example, findings from a two-wave longitudinal study among 179 firefighters suggest that humor is a coping mechanism that buffers the negative effects of being exposed to traumatic stressors at work [32]. Moreover, a survey study with 73 community-dwelling older adults revealed that humor as a coping stress mechanism was associated with mental health status [33]. Similarly, Fritz and colleagues [34] conducted three studies, comprising both fibromyalgia patients and undergraduate students, that examined the relationship between humor and adjustment to stressful events. They concluded that “one way humor decreases distress is by facilitating positive reframing of the event” (p. 852). In the organizational context, findings from three studies with different methodologies, from a lab experiment to a three-wave survey study, offer some casual evidence of this buffering effect: employees’ use of stress-relieving humor helps them to both reduce stress appraisals and cope better with the stressor [35]. The explanation of these results is that “humor makes light of a stressor or stressful situation and helps people feel better and think more positively about a stressful situation” [35] (p. 4).

Second, from a physiological perspective, humor elicits laughter, which releases endorphins and relieves stress [32]. Two recent meta-analyses in healthcare settings, one examining 10 randomized control trials (RCTs) comprising 814 adults and another analyzing 29 RCTs and quasi-experimental studies comprising 1986 participants, concluded that laughter-inducing therapies and humor interventions reduced depression and anxiety [36,37]. Overall, these meta-analyses reported small to large between-group effect sizes for depression and medium to large between-group effect sizes for anxiety.

Finally, considering an interpersonal approach, establishing good relationship with coworkers and perceiving that you can count on their support is considered both a pivotal resource to deal with work stress and a source of well-being. In work settings, some studies have shown that humor can aim at enhancing social connection and ties among people (i.e., affiliative humor), which improves workplace relationships, resulting in leaders and groups that manage better their emotions and are more productive [38,39,40,41,42,43]. Such social bonds can translate into a source of social support that buffers the negative effects of work stress [44]. Moreover, within-subjects analyses from a diary study of 57 Dutch workers showed that employees reported being less emotionally exhausted and more engaged on days when they expressed self-enhancing and affiliative forms of humor [45].

Taking together the available evidence on the relationship between humor and well-being and considering that a positive and active humor can be trained (for a review, see [23]), we examine the effectiveness of a training program aimed at promoting an adaptive humor at work and reducing psychological distress.

## 2. Materials and Methods

### 2.1. Procedure and Participants

The study was conducted in an emergency ambulance service in Andalusia (Spain). This study was part of the voluntary training that the organization offers to their employees each year. Participants received a welcome email explaining the schedule and content of the training. Instructions to participate in the study were given in this email. No specific selection criteria were applied to participate in the training, so we studied a convenience sample. According to the American Psychological Association’s (APA) Ethical Principles of Psychologists and Code of Conduct, participants were informed about the aim of the study and the requisites for voluntary and confidential participation, and all participants gave their written informed consent.

Of the 62 participants enrolled in the training (in two groups of 35 and 27 participants), 58 accepted to participate in the study, whereas 4 declined to participate because they considered that they would not be able to meet all the requirements of the training program. We used a one-group pre–post training research design. Thus, all participants received the training during their working schedule over six weeks (two 90 min sessions per week plus a follow-up session one month later, in which participants evaluated the training with the person responsible for the training and capacitation area of the company). Data collection was undertaken in the workplaces during the training sessions in the presence of the last author, who was also the trainer. Participants placed their completed questionnaires in a sealed box to ensure the anonymity of their responses. Their responses were matched with a code that they created (last four digits of their mobile phone plus four digits with the day and month of their birth).

Fifty-eight workers participated in the study (46.6% women and 53.4% men), most of them were medical staff (52% nurses and doctors vs. 48% technicians and support staff) with a permanent contract (79.3% vs. 20.7% temporal contract) and experience in their jobs (*M* job tenure = 13.68 years; *SD* = 6.67). Their age ranged between 27 and 57 years (*M* = 42.72; *SD* = 7.81).

Regarding statistical analyses, all analyses were performed using IBM SPSS^®^ version 26, except the analysis for establishing the statistical power of the study, which was carried out with G*Power 3.1.

### 2.2. Rationale and Content of the Training

Regarding the training, we followed both the theoretical approach that differentiates between an adaptive (affiliative and self-enhancing) and a maladaptive (aggression and self-defeating) use of humor [44,45] and the humor components described by Thorson and Powell [17,18]: humor generation or creativity, humor appreciation and tolerance of ambiguity, use of humor to achieve social goals, and humor as an adaptive or coping mechanism. Therefore, we focused on training skills and humor elements that make more likely using adaptive humor to maintain positive social interactions and deal with stressful situations. In particular, the training consisted of 13 sessions lasting 90 min each.

We used a structured learning strategy similar to that employed in social skills training procedures, that is, giving instructions on the dimension of humor that is going to be learned, exemplifying it through behavioral models, encouraging participants to practice following the role model provided, giving feedback on their execution, and providing guidelines for the generalization of what was learned in class to their job context [46,47,48,49]. In that sense, the in-class teaching methodology or approach was participatory, including exchanges among participants to identify appropriate role models and performing role-playing techniques to practice their implementation in different work situations.

The content of the training was inspired by both Garcia-Larrauri’s [21] and Ruch and colleagues’ [23,24] programs for improving the sense of humor (see Table 1). First, participants attended an introductory session aimed at understanding humor from a scientific perspective and the benefits of using positive or adaptive humor at work versus other types of humor such as sarcasm, mockery, or offensive jokes (session 1). Then, during the main sessions, most of the activities were aimed at enhancing the elements and skills of humor, such as exploring the factors that facilitate good humor (e.g., being tolerant with oneself, encouraging wit and creativity, using positive language, or smiling) and the use of positive humor at work (session 2); increasing appreciation of humor by sharing stories with incongruous and funny incidents and making jokes out of mistakes made by oneself (i.e., encouraging wit and self-tolerance) (sessions 3–4); enhancing creativity and flexible thinking to intentionally create positive humor, which included activities such as connecting ideas through the question “What happens if…?” or telling stories with randomly selected words (sessions 5–7); anticipating adverse conditions at work and coping with adversity and stress in a more flexible and positive way, for example by recalling episodes from one’s own biography or constructing stories that involved answering yes to the following question: “Is a life without problems worth living?”, or contrasting the approach used to deal with a past problem with a futuristic perspective, or dramatizing feared work situations to the point of absurdity (i.e., using positive humor under stressful situations) (sessions 8–10); developing an assertive communication style in which positive humor can play a pivotal role (sessions 11–12). Finally, there was a follow-up session one month later (session 13), in which the person responsible for the training and capacitation area of the company conducted post-training data collection and then group discussions to monitor the training and explore the difficulties that trainees experienced in applying the skills learned to their work settings.

### 2.3. Measures

*Sense of humor.* This variable was measured by the Multidimensional Sense of Humor Scale (MSHS) [17,18], which contains 24 statements, 18 of which are positively worded (e.g., “I appreciate those who generate humor”) and 6 are negatively worded (e.g., “People who tell jokes are a pain in the neck”) to reduce the fixed response bias. The response scale ranges from 0 (*strongly disagree*) to 4 (*strongly agree*). Negatively worded items are recoded (inversed scores) to facilitate the interpretation of the scores. We used the Spanish version of the scale whose translation was supervised by its creators [50]. The Cronbach alpha coefficient of the scale was 0.82 (pre-training) and 0.83 (post-training).

Given the number of participants, no factor analysis of the scale was performed, assuming the factor structure proposed by the original authors [17,18]. Thus, the 24 items were divided into 4 subscales or factors: (1) *Humor generation*. Composed of 11 items that measure a person’s confidence in her/his ability to make others laugh with witticisms and inventiveness (e.g., “I’m confident that I can make other people laugh”, “My clever sayings amuse others”). The Cronbach alpha coefficient of this dimension in this sample was 0.88 (pre-training) and 0.67 (post-training); (2) *Coping through humor*. Composed of 7 items measuring the ability to use humor as a coping mechanism (e.g., “I can use wit to help adapt to many situations”) and a way of coping with difficult situations (e.g., “Uses of wit or humor help me master difficult situations”). The Cronbach alpha coefficient of this dimension in this sample was 0.48 (pre-training) and 0.53 (post-training); (3) *Appreciation of humor.* Consists of 2 items that measure the recognition and merit that a person gives to those who can generate humor (“I appreciate those who generate humor”) and to humor itself (“I like a good joke”). The Cronbach alpha coefficient of this dimension in this sample was 0.83 (pre-training) and 0.46 (post-training); and (4) *Attitude toward humor*. Consists of 4 negatively formulated items that, when inverted, measure the favorable disposition towards humor and fun (e.g., “Calling somebody a “comedian” is a real insult”). The Cronbach alpha coefficient of this dimension in this sample was 0.88 (pre-training) and 0.47 (post-training).

*Cheerfulness and seriousness* were measured as ‘states during the last week’ with the State-Trait Cheerfulness Inventory (STCI-S) in its Chilean Spanish version [51]. These two dimensions of the STCI address the prevalence of a cheerful/serious mood and the perception of daily events according to such mood. We did not include the dimension ‘bad mood’. We used 20 self-report items and a 4-point Likert scale ranging from 1 (‘strongly disagree’) to 4 (‘strongly agree’) for measuring cheerfulness (e.g., “I was cheerful”, “I saw the funny side of things”; αpre = 0.83; αpost = 0.85), and another 20 items to measure seriousness (e.g., “I was set for serious things”, “I was in a sober frame of mind”; αpre= 0.75; αpost= 0.79). Higher scores indicate having experienced each state in a higher degree during the last week (total score 20–80).

*Psychological distress* was measured by the Spanish version of the General Health Questionnaire in its 28-item version (GHQ-28) [52]. Each item has four answering categories: better than usual, same as usual, worse than usual, and much worse than usual. We used the common scoring method that attributes a binary score system of 0 to the first and second response options (better than usual, same as usual) and a score of 1 to the third and fourth response options (worse than usual, much worse than usual). Therefore, higher scores indicate higher levels of self-reported psychological distress. Indeed, this version provides both a total score (i.e., psychological distress, ranging from 0 to 28) and four scores measuring somatic symptoms, anxiety and insomnia, social dysfunction, and severe depression (ranging from 0 to 7). The Cronbach’s alpha coefficient of the scale in this sample was 0.93 (pre-training) and 0.92 (post-training). Regarding the subscales, the following coefficients were obtained: somatic symptoms 0.82 (pre-training) and 0.75 (post-training); anxiety and insomnia 0.90 (pre-training) and 0.73 (post-training); social dysfunction 0.79 (pre-training) and 0.74 (post-training); and depression 0.98 (pre-training) and 0.74 (post-training).

## 3. Results

First, we calculated descriptive statistics and bivariate correlations between the main variables of our study (see Table 2). We also tested for differences in the main variables depending on the sociodemographic characteristics of the sample. Overall, one-factor ANOVA analyses did not reveal any significant difference in the scores of the main variables depending on sex (0: male vs. 1: female), contract type (0: permanent vs. 1: temporary), job category (0: nurses and doctors vs. 1: technicians), or training group. An exception was that people with a permanent contract reported a significantly higher mean on the pre-training measure of humor generation (*M* = 2.59; *SD* = 0.53) than those whose contract was temporary (*M* = 2.18; *SD* = 0.78) (*t*(56) = 2.1; *p* < 0.05: 95% CI [0.019; 0.783]).

Regarding the effectiveness of the training, we first performed Kolmogorov–Smirnov (K-S) tests to check the normality assumption necessary to conduct paired-samples t-test to compare changes in the main variables before and after the training. The results revealed that the distributions of most variables did not fit the normal curve (*p* < 0.05), so we opted for performing Wilcoxon signed-rank tests instead of paired-samples t-tests to assess the effectiveness of the training. The results indicated, in line with expectations, that after training the scores were higher in positive attitudes toward humor and cheerfulness and lower in seriousness, psychological distress, and all its dimensions except for somatic complaints (see Table 3 and Figure 1 and Figure 2), but contrary to expectations, there were lower scores in humor appreciation and higher scores in social dysfunction symptoms after the training (see Table 3). The same results were found when parametric tests (paired samples t-tests) were used instead of non-parametric tests (Wilcoxon signed-rank tests).

We also calculated the effect size for nonparametric data by dividing the z value by the squared root of N [53], where N refers to the number of observations over the two time points (in our case, N = 58 participants × 2 times = 116). The results indicated a medium effect size of our humor-based training program for cheerfulness (r = 0.36), seriousness (r = 0.31), and psychological distress (r = 0.31) and a large effect size for both appreciation of humor (r = 0.51) and attitudes toward humor (r = 0.44). Furthermore, as this study was based on a convenience sample, we performed post hoc power analysis for Wilcoxon signed-rank test (matched pairs) to calculate the statistical power of the study. We introduced an effect size of 0.5, an alpha error probability of 0.05, and a sample size of 58. The results revealed a statistical power of 0.99 (1-β error probability), suggesting that there is a high probability that our training had an effect (i.e., there is low probability of making a Type II error).

In addition, after training, we observed a 10.4% reduction of potential cases of minor psychiatric disorder when applying a cut-off score of 6 points or higher in the psychological distress measure (GHQ) [52]. In other words, using such cut-off score led to the finding that 62.1% of our sample was at high risk of developing a minor psychiatric disorder before the training. After training, people at high risk were reduced to 51.7%. A more conservative cut-off score of 7 or higher was also associated with a 5.2% decrease in people at high risk of developing a minor psychiatric disorder after training (pre-training = 55.2% vs. post-training = 50%).

## 4. Discussion

Healthcare workers use humor for different purposes, such as to relieve tension in difficult situations that challenge their professional abilities, be more benevolent with their own and others’ mistakes, manage patients’ negative emotions and fear and provide a high-quality service, maintain composure in grotesque situations, foster supportive relationships among coworkers [54,55,56], and recently, cope with the stress caused by work overload during the COVID-19 pandemic and the distress caused by the high risk of becoming infected [57,58].

However, previous evidence has being limited to the qualitative analysis of the spontaneous use of humor in healthcare professionals working under pressure and/or in hospital emergency units, and, to our knowledge, no previous study had tested the effectiveness of a training program aimed at improving an adaptive humor at work and reducing stress in emergencies workers [54,55,56]. Indeed, few studies have addressed humor-based stress reduction interventions in organizational settings, with the only exception of a study that evaluated a short-term intervention with aspiring nurses [22].

In response, this study evaluated the effectiveness of a training program aimed at promoting an adaptive humor at work as a mechanism to deal with daily workplace stress in emergency healthcare professionals.

As expected, our results provide evidence of the suitability of promoting humor at work to reduce the effects of work stress and achieve greater psychological well-being in healthcare workers. In this regard, psychological distress scores were significantly reduced after the training, particularly, anxiety and depression symptoms, which seems to indicate that the training intervention contributed to the psychological well-being of the professionals who participated in it. Regarding the potential mechanisms that may help explaining these results, our results show that after the training, the state of cheerfulness increased and the state of seriousness decreased. In other words, after training, the temporary disposition to use humor to cope with difficulties and stressful situations at work increased, which could have helped workers to relativize the severity of these stressful situations. Moreover, while the attitude towards humor increased from pre-test to posttest, the appreciation of humor decreased. In this respect, as well as the fact that these scales have lower reliability coefficients in the post-test, we consider that a possible explanation is that after training, the participants learn to differentiate between different ways of using humor and to discriminate which of these they prefer or consider to be appropriate or not in their work, so that they respond inconsistently to the statements of these dimensions, even increasing their willingness to use humor.

In addition, the effectiveness of our humor-based intervention to reduce psychological distress can be explained from both a cognitive and a physiological perspective. In that sense, the ability to generate and using humor to adjust to certain work situations presupposes the elaboration of perceptual and cognitive schemes that guide action or behavior, which is common to the cognitive–behavioral approach that has amply demonstrated its ability to support effective interventions for stress management [9]. Furthermore, humor leads to laughter with vehemence and unbalanced movements that release tension and stress [36,37]. In other words, the laugh that usually accompanies humor presupposes a certain state of relaxation, and the procedures devised to induce relaxation and control work stress also have solid evidence of its effectiveness [10]. Promoting an adaptive use of humor and related behaviors that denote joviality and gentleness may also facilitate interpersonal relationships at work, group coordination, and better communication with patients to achieve their collaboration [39,40,41,42], in addition to being a source of social support that moderates the negative effects of work stress [43,59,60,61].

In a nutshell, in line with transactional stress theories, our findings allow us to understand humor as a coping mechanism that can be developed through training and whose function would be to reduce, cancel, or reverse the effect of stressful work situations [62].

Although our results are promising, our study design has some limitations that need to be overcome in the future. The design used (pre-test–post-test of a single group) is pre-experimental and, therefore, it cannot be assured that the changes observed are solely the result of training. Among the threats to the internal validity of this study, the most plausible would be the history or time lag between pre-test and post-test, repeated performance of the same tests, reactivity to the measurement process, and statistical regression to the mean. Along with training, there may be other external variables over which there is no absolute control, which makes it difficult to rule out alternative explanations for our results. In other words, the training was implemented during some months, and other factors beyond the training could have affected our results. In that sense, although the experimental procedure was the same for all the participants and we controlled for their sex, job category, and contract type, it is possible that stressors may have varied during training implementation and changed between the measurement points (before and after the training), resulting in workers exposed to different work pressures and stressful situations. In this regard, although natural settings are complex, future studies should follow a randomized controlled trial design in which participants are randomized to either the intervention or the control group. As participants’ randomization is not always possible, we encourage to include a comparison group to provide a more robust test of the effects of the training.

In addition, more objective indicators beyond self-reported measures should be considered to assess the effectiveness of the training. For example, salivary cortisol levels can be used to measure stress and external evaluations, or peer-to-peer reports can be used to measure adaptive humor. Moreover, future research should incorporate measures with better psychometric properties for assessing participants’ sense of humor and its possible dimensions. In particular, the dimensions of the scale MSHS presented low internal consistencies, while there was a strong intercorrelation among them. This result suggests both that focusing on just the total score of the scale can be more valid than describing the results according to the scores in each dimension and that further studies should analyze the factorial solution of the scale in different cultural contexts.

Besides methodological issues, we opted for describing the use of humor as either adaptive or maladaptive in terms of the nature of the humor content. However, from a contingent perspective, there is no a priori use of humor that is more adaptive as it depends on contextual variables. Thus, future studies may find useful to describe humor by other elements rather than its potential outcomes.

Finally, future studies should include measures of potential variables that may explain the role of humor as a stress-coping mechanism. For example, future studies can measure potential mediators or mechanisms, such as group cohesion and social support or stressors’ appraisals as challenges or threats, that can contribute to explain the relationship between humor and stress at work. Furthermore, combining external reports about the use of humor with physiological indicators may contribute to clarifying how humor triggers stress relief responses.

## 5. Conclusions

Despite the limitations inherent to our design and the lack of reliability of the post-test measures of some dimensions of the Multidimensional Sense of Humor Scale, our findings have interesting practical implications for developing healthy and positive organizations. This study contributes to the existing literature by providing a concrete guideline to train healthcare workers in promoting humor at work as a stress-coping mechanism, which extends the application of humor-based programs that had already demonstrated their effectiveness in other sectors [22,23,24,25,26,27].

Indeed, our findings can help to shed some light on the current debate about when positive interventions are beneficial [63]. In line with a Positive Organizational Psychology perspective [64,65], our results revealed that our humor-based training may be useful for healthcare workers to manage work stress. This is particularly interesting because healthcare workers are one of the professional groups most exposed to work stress and its adverse effects, before and after the current COVID-19 crisis [7,8]. Indeed, research has shown that using humor is a good way to deal with work stress, so it is recommended to include it as a coping tool in wider programs for managing work stress.

In conclusion, our training seems to be a potential alternative or complement to more traditional workplace stress management interventions.

## Figures and Tables

**Figure 1 ijerph-18-11177-f001:**
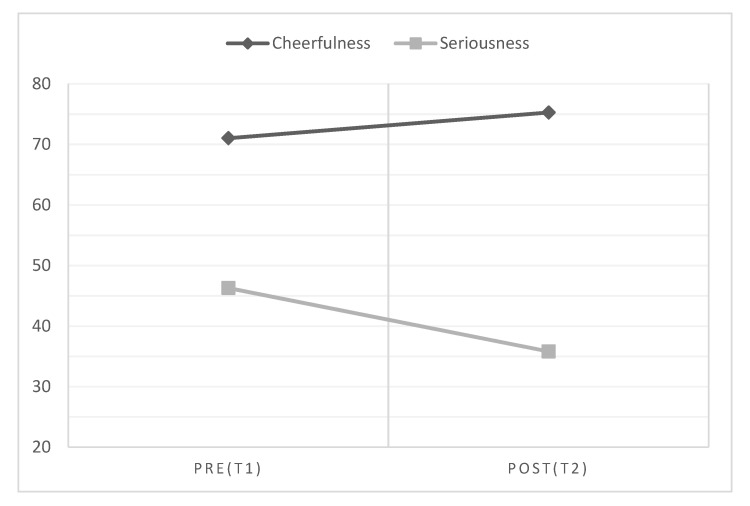
Cheerfulness and seriousness scores both before and after training (N = 58). For clarity, the *Y*-axis shows mean scores (not mean ranks).

**Figure 2 ijerph-18-11177-f002:**
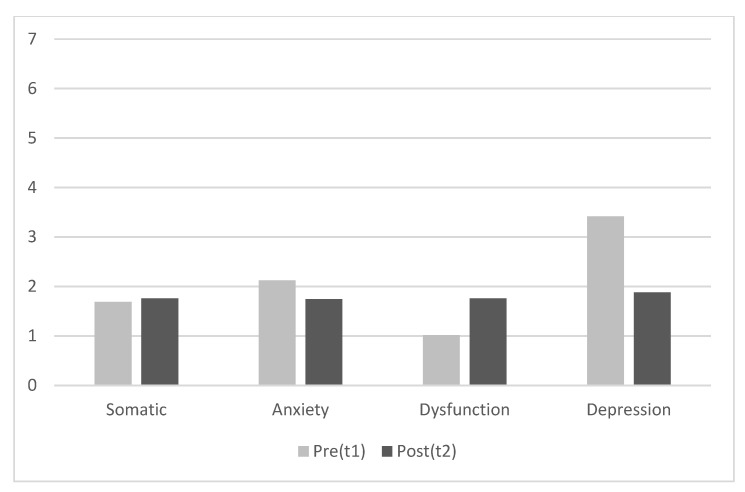
Scores for psychological distress dimensions both before and after training (N = 58). For clarity, the *Y*-axis shows mean scores (not mean ranks).

**Table 1 ijerph-18-11177-t001:** Contents of the training.

Content	Sessions	Objectives and Activities
Welcome	0	Sending welcome email with training schedule and a participation letter that explained the study.
Introduction	1	Conducting pre-training data collection. Understanding the sense of humor from a scientific perspective and the benefits of using positive humor at work (vs. other types of humor).
Humor appreciation	2–4	Exploring the factors that facilitate good mood and the use of positive humor at work. Being more tolerant with oneself.
Humor creation	5–7	Improving creativity and flexible thinking in order to intentionally create positive humor.
Humor as coping	8–10	Anticipating adverse conditions at work and coping with adversity in a more flexible and positive way. How to use positive humor under stressful situations.
Positive exchanges	11–12	Developing an assertive communication style in which positive humor can play a pivotal role.
Follow-up	13	Monitoring the training and conducting post-training data collection.

**Table 2 ijerph-18-11177-t002:** Means, standard deviations, and bivariate correlations among the main variables of the study (*n* = 58).

Variables			Post-Training											
Pre-Training	Mean ^a^	SD ^a^	1	2	3	4	5	6	7	8	9	10	11	12
1. Sense of humor	2.43	0.47	0.90 **	0.82 **	0.72 **	0.58 **	0.62 **	0.21	0.46 **	−0.39 **	−0.34 **	−0.39 **	−0.46 **	−0.21
2. Humor generation	2.50	0.61	0.59 **	0.53 **	0.52 **	0.31 *	0.42 **	−0.13	0.47 **	−0.22	−0.12	−0.26	−0.26 *	−0.15
3. Coping through humor	2.53	0.54	0.72 **	0.67 **	0.58 **	0.43 **	0.50 **	0.23	0.24	−0.28 *	−0.28 *	−0.30 *	−0.37 **	−0.05
4. Appreciation of humor	3.50	0.84	0.49 **	0.46 **	0.37 **	0.33 *	0.33 *	−0.14	0.38 **	−0.18	−0.11	−0.20	−0.27 *	−0.04
5. Attit. towards humor	1.52	1.28	0.52 **	0.49 **	0.35 **	0.46 **	0.35 **	0.50 **	0.11	−0.31 *	−0.35 **	−0.25	−0.30 *	−0.22
6. Seriousness	46.29	21.44	0.47 **	0.45 **	0.29 *	0.43 **	0.30 *	0.55 **	0.07	−0.28 *	−0.33 *	−0.21	−0.27 *	−0.17
7. Cheerfulness	71.03	8.34	0.60 **	0.54 **	0.52 **	0.33 *	0.42 **	−0.18	0.45 **	−0.25	−0.16	−0.31 *	−0.28 *	−0.13
8. Psychol. distress (PD)	8.24	6.26	−0.39 **	−0.30 *	−0.33 *	−0.43 **	−0.27 *	0.09	−0.12	0.92 **	0.81 **	0.80 **	0.82 **	0.82 **
9. PD-Somatic symptoms	1.69	1.99	−0.24	−0.16	−0.27 *	−0.24	−0.12	0.29*	−0.11	0.68 **	0.56 **	0.62 **	0.66 **	0.55 **
10. PD-Anxiety/insomnia	2.12	2.29	−0.29 *	−0.21	−0.24	−0.37 **	−0.20	0.11	−0.18	0.81 **	0.72 **	0.78 **	0.70 **	0.67 **
11. PD-Social dysfunction	1.01	1.61	0.08	0.13	0.08	−0.11	0.02	0.01	0.08	0.34 *	0.22	0.31 *	0.21	0.47 **
12. PD-Depression	3.41	3.19	−0.45 **	−0.39 **	−0.34 **	−0.37 **	−0.32 *	−0.10	−0.07	0.62 **	0.61 **	0.47 **	0.59 **	0.54 **
Mean ^b^	-	-	2.47	2.50	2.43	2.36	2.49	35.81	75.28	7.14	1.76	1.74	1.76	1.88
SD ^b^	-	-	0.56	0.61	0.63	1.08	0.80	12.15	4.19	6.09	1.76	1.72	1.66	1.72

Note: * *p* < 0.05; ** *p* < 0.01; ^a^ Mean and standard deviation at the pre-training measurement point; ^b^ Mean and standard deviation at the post-training measurement point.

**Table 3 ijerph-18-11177-t003:** Results from Wilcoxon Signed-Ranks tests (post–pre training measures).

Variable	Mean Rank Positives	Mean Rank Negatives	*Z*	*p*	*r*
Sense of humor	30.63	25.28	−0.73	0.462	0.07
Generation of humor	25.52	29.98	−0.20	0.843	0.02
Coping through humor	21.71	29.81	−1.33	0.182	0.12
Appreciation of humor	10.50	25.26	−5.55	0.001	0.51
Attitudes toward humor	33.19	12.70	−4.76	0.001	0.44
Cheerfulness	30.50	13.66	−3.93	0.001	0.36
Seriousness	15.50	36.40	−3.32	0.001	0.31
Psychol. Distress (PD)	18.60	22.41	−3.35	0.001	0.31
PD-somatic complaints	22.23	20.70	−0.48	0.634	0.04
PD-anxiety and insomnia	16.46	18.80	−1.96	0.050	0.18
PD-social dysfunction	22.91	21.27	−3.09	0.002	0.29
PD-depression	12.18	24.23	−3.86	0.001	0.36

## Data Availability

The data presented in this study are openly available on the website Open Science Framework: https://osf.io/cp5hv/files/ (accessed on 20 October 2021). In the manuscript, we also report how we determined sample size, all measures in the study, all data exclusions, and all manipulations.

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
