# Peer review of "Effectiveness of a Humor-Based Training for Reducing Employees’ Distress"

_ijerph, 2021, doi:10.3390/ijerph182111177_

Round 1

Reviewer 1 Report

The authors are right that there are few published studies on humor interventions at the work place. Even more so, the theoretical foundations are not clear yet as findings from humor studies are not incorporated. There are many groups involved in humor research; some actually studying humor and other using humor in their respective fields. I am glad to see a humor training study at the work place and so this article indeed fills a gap.

There are two comments on conceptual issues that are necessary, namely on the use of the MSHS and the HSQ variables. The use of both creates an apparent inconsistency. The sense of humor was usually seen as a unidimensional concept measuring the quantity of humor. “Sytyles” of humor represents an alternative and incompatible approach looking at qualitative differences. And both are flawed making the present study stand on weaker foundations. I will discuss this next.

1) You write “Sense of humor. This variable was measured by the Multidimensional Sense of Humor 144 Scale (MSHS) [39], which contains 24 statements”. In the table I see ONE total score of sense of humor. So a “multidimensional” scale yields only ONE total score? How is this to be understood? If the scale is multidimensional then I would expect a profile of humor, i.e., more than one result per person. If a total score is used, but more contents are covered, then this is a very heterogeneous (and hence invalid) scale. This is a scale not to be used in research due to this inconsistency. What you can do is use the factors they found as separate scores. Maybe you get better results for this.

You also wrote: “On the other hand, results revealed non-significant changes on sense of humor after training (see Table 3).” This is highly surprising as there are changes in all other variables, and cheerfulness and low seriousness will be highly correlated with the sense of humor. One can also see that the Sense of humor does not have a pre post stability (.22) but all the other variables have. Is there a mistake in computing the scores? Or does it reflect the heterogeneity of the items (as indicated by “multidimensional”) and the fact that some of these components change while others don’t. Can you wrote something to diffuse my concerns? If not then add the use of the MSHS and what consequences it had to the limitations.

2) You write “Regarding the training, we followed both the distinction of humor styles that differentiate between adaptive (affiliative and self-enhancing) and maladaptive (aggression 102 and self-defeating) styles…”. The “adaptive” vs. “maladaptive” humor distinction is clouding the thinking of humor researchers for almost 20 years now and this is very unfortunate. While certain forms of humor may be adaptive and others maladaptive this is not a useful way of describing different forms of humor. Whether or not using forms of humor is adaptive is a matter of empirical research, but it should not be used to describe humor as it explains or describes nothing. For example, telling jokes and funny stories (a core of affiliative humor) maybe highly dysfunctional; e.g., when you tell racist of otherwise inappropriate jokes to certain audiences.  Aggressive humor, in the form of satire, is highly adaptive if you want to undermine the status of a dictator. It is better to describe the humor by what it really is.. and not by one potential outcome. See, for example, https://www.degruyter.com/document/doi/10.1515/humor-2017-0089/htmlhttps://www.degruyter.com/document/doi/10.1515/humor-2015-0095/html https://econtent.hogrefe.com/doi/10.1027/1015-5759/a000440 I think you should discuss the strengths and limitations of this approach so that readers of your article don’t fall into the same trap.

Minor issues:

Maybe add a few thoughts on the importance of using comparison groups and what they might be. Such as placebo control group or, alternative training group.

You write “Humor can be defined as “amusing communications that may generate positive feelings and cognitions in employees” .  I would rather quote authoritative works than peripheral work when it comes to such fundamental question of what humor is. In other words, as an author you have the responsibility to guide the reader through the literature and point the to solid domains. 

Author Response

Dear Reviewer # 1,

Thank you for your time and consideration. Following your suggestions, we have now submitted an improved version of our manuscript. In particular, we have addressed your comments as follows:

Main concerns:

  1. Thank for your comment and raising this issue. We followed the rationale of providing an overall score because the factorial configuration has not been tested in our context and the small sample of our study prevents us from conducting a confirmatory factor analysis. However, we agree with your concerns, and we have followed the original dimensions proposed by Thorson and Powell (see measures section, lines 195-220). Thus, we have conducted again the analysis to include the dimensions of the scale (see results section and some sentences in the discussion, such as: “In this respect, as well as the fact that these scales have lower reliability coefficients in the post-test, we consider that a possible explanation is that after training the participants learn to differentiate between different ways of using humor and to discriminate which of these they prefer or consider to be appropriate or not in their work, so that they respond inconsistently to the statements of these dimensions, even increasing their willingness to use humor.”). Also, as the scale has some limitations regarding its psychometric properties, we have included them in the discussion (lines 377-378: “Moreover, future research should incorporate measures with better psychometric properties for assessing sense of humor and its possible dimensions.”)

  1. Thank for your time and suggestions. We agree that the definition of humor was not clear enough. In response, we have described the perspective that guided the study in the lines 55 to 71 (introduction). Also, we have checked the text to be more accurate in our expressions. For example, we have opted for “use of humor” rather than “humor style”. Finally, we have included the following sentence in the limitations: “Besides methodological issues, we opted for describing use of humor as either adaptive or maladaptive in terms of the nature of the humor content. However, from a contingent perspective, there is no a priori use of humor that is more adaptive as it depends on contextual variables. Thus, future studies may find useful to describe humor by other elements rather than its potential outcomes.”

Minor issues:

  1. Thank you for your comments. We have further elaborated the limitations and included some thoughts about the importance of having a control group: “The design used (Pretest-Posttest of a single group) is pre-experimental and, therefore, it cannot be assured that the changes observed are solely the result of training. Among the threats to the internal validity of this study, the most plausible would be the history or time lag between pretest and posttest, repeated performance of the same tests, reactivity to the measurement process and statistical regression to the mean. Along with training, there may be other external variables over which there is no absolute control, which make difficult to rule out alternative explanations for our results. In other words, the training was implemented during some months and other factors beyond the training can affect or results. In that sense, although the experimental procedure was the same for all the participants and we controlled for their sex, job category and contract type, it is possible that stressors may vary during the training implementation and changed between measurement points (before and after the training), resulting in workers exposed to different work pressures and stressful situations. In this regard, although natural settings are complex, future studies should follow a randomized controlled trial design in which participants are randomized to either the intervention or the control group. As participants’ randomization is not always possible, we encourage to include a comparison group to provide a more robust test of the effects of the training.“

Reviewer 2 Report

Review for  ijerph-1369227 entitled “Effectiveness of an Adaptive Humor Training for Reducing Employees’ Distress”.

The manuscript describes changes in sense of humor, cheerfulness, seriousness, and stress-related variables following a humor intervention at the workplace in a sample of N = 58 employees. The authors conclude that their intervention was an effective intervention for managing work stress.

The manuscript is well-written and interesting to read. However, given the study design, the authors need to describe their results, and mostly, their interpretations more carefully.

Major points:

  • My biggest point of concern throughout the manuscript is the lack of a comparison condition. Without this, it is not possible to determine whether the intervention had an effect, or whether the results are based on external factors, placebo effects, or similar. This limitation must be considered throughout the manuscript. Thus, please do not write sentences like “the intervention increased” (but rather, for example: cheerfulness increased following the intervention) and do not speak about the “effectiveness” of the intervention – intervention effectiveness cannot be studied in this design. Along these lines, I recommend framing the study as a pilot study
  • Introduction: “Humor can be defined as…” – here the authors are referring to the role of humor at work – not humor in general, please revise.
  • Methods:
    • Given the authors’ origin, I assume the study was conducted in a Spanish sample. Please add this information to the manuscript. Also, I assume that Spanish versions of the instruments were used. Please provide references to the Spanish versions of the manuscript or describe the translation procedures.
    • Please describe your considerations for statistical power. E.g., what effect size did the present sample size allow you to detect?
    • Were there no dropouts?
    • Please include the following statement (if applicable):
      “We report how we determined our sample size, all data exclusions (if any), all manipulations, and all measures in the study”
    • Does the multidimensional sense of humor scale allow for computing a total score? Also, I was not sure about what this scale measures. I recommend providing a description/definition of “sense of humor” how it is used in this measure.
    • GHQ: Since the authors report results on both, the level of the total score and the subscales, they should provide estimates of internal consistencies for both.
  • Results
    • The means in sense of humour actually decrease following the intervention. This should be mentioned and discussed.
    • Figures 1 and 2: Please mention that the y-axis shows mean scores (while the statistical analyses is based on mean ranks). Also, I recommend also giving confidence intervals.

Author Response

Dear Reviewer # 2,

Thank you for your time and consideration. Following your suggestions, we have now submitted an improved version of our manuscript. In particular, we have addressed your comments as follows:

  1. Thank you for raising this issue about the lack of comparison group. We concur that not having a control group is an important limitation. However, we cannot frame our study as a pilot study as it is an intervention with a weak design, but it is not a pilot study. The same happens with cross-sectional studies compared to longitudinal studies. One design is more robust than other from a methodological point of view, but it does not mean that any of them is invalid or incorrect. Of course, we have followed your advice and we have paid attention not to make causal inferences. Also, we have added the following paragraph in the limitations: “The design used (Pretest-Posttest of a single group) is pre-experimental and, therefore, it cannot be assured that the changes observed are solely the result of training. Among the threats to the internal validity of this study, the most plausible would be the history or time lag between pretest and posttest, repeated performance of the same tests, reactivity to the measurement process and statistical regression to the mean. Along with training, there may be other external variables over which there is no absolute control, which make difficult to rule out alternative explanations for our results. In other words, the training was implemented during some months and other factors beyond the training can affect or results. In that sense, although the experimental procedure was the same for all the participants and we controlled for their sex, job category and contract type, it is possible that stressors may vary during the training implementation and changed between measurement points (before and after the training), resulting in workers exposed to different work pressures and stressful situations. In this regard, although natural settings are complex, future studies should follow a randomized controlled trial design in which participants are randomized to either the intervention or the control group. As participants’ randomization is not always possible, we encourage to include a comparison group to provide a more robust test of the effects of the training.”

  1. We have added information about the Spanish version of the instruments used in our study. We have included effect sizes, information about dropouts, and estimates of internal consistency of the total scale and its dimensions in the case of sense of humor and psychological distress. In addition, we have provided more information about the sense of humor scale (see response to reviewer #1).

  1. We have followed your recommendations and we discuss both expected and unexpected results. Also, we have included a note in the figures to mention that “For clarity, the Y-axis shows mean scores (not mean ranks).”

Round 2

Reviewer 2 Report

Dear authors,

Thank you for considering my suggestions in the revision of the manuscript.

  • Two of my previous comments were not considered:
    • Please describe your considerations for statistical power, i.e., how did you determine the sample size?
    • In line with efforts to increase research transparency, I recommended to include the following statement (see http://dx.doi.org/10.2139/ssrn.2160588):
      “We report how we determined our sample size, all data exclusions (if any), all manipulations, and all measures in the study”
  • Further, some new comments came up when going through the manuscript again:
    • Given the very low internal consistencies of the MSHS subscales and the strong intercorrelations among its subscales, I recommend only describing and reporting the total score, given that the scale does allow for calculating such a total score.
    • The numbering of references is wrong (e.g., 19, 20 are not refernces for the MSHS)

Author Response

Dear Reviewer #2,

Thank you for your time and valuable comments and suggestions. We have added your comments in our manuscript as follows:

1.- It is great to see how reviewers are in the same line than us. We are also committed with a more transparent and open science. Thus, we have highlighted that the sample is a convenience sample (see line 127) and therefore we did not determine the sample size in advance, something that is highly recommended for establishing the statistical power of the study. Accordingly, we have performed post hoc analysis of statistical power with G*Power (see p. 4, lines 149-151: “Regarding statistical analyses, all analyses were performed using IBM SPSS® version 26, except the analysis for establishing the statistical power of the study, which was determined with G*Power 3.1.”). Also, we report the results of such power analysis: “Furthermore, as this study is based on a convenience sample, we performed post hoc power analysis for Wilcoxon signed-rank test (matched pairs) to calculate the statistical power of the study. We introduced an effect size of 0.5, an alpha error probability of 0.05, and a sample size of 58. Results revealed a statistical power of 0.99 (1-β error probability), suggesting that there is a high probability that our training had an effect (i.e., there is low probability of making a Type II error).” (see p. 7, lines 284-290).

2.- Thank you for your suggestion. We have added such comment in the “data availability comment” at the end of our manuscript: “The data presented in this study are openly available in the website Open Science Framework: https://osf.io/cp5hv/files/. Also, in the manuscript, we report how we determined or sample size, all measures in the study, all data exclusions, and all manipulations.”

3.- Thank you for raising this issue. Indeed, that was the reason why we decided to present only the total score of the MSHS scale in the first version of the manuscript. We are sorry that we did not clarify this issue enough (and then the scores of all dimensions should have been included as supplementary analysis). Now, we think that is better keeping the manuscript as it is, which is in line with reviewer’s #1 suggestion of including all the information. However, we have highlighted this issue in the discussion: “Moreover, future research should incorporate measures with better psychometric properties for assessing sense of humor and its possible dimensions. In particular, the dimensions of the scale MSHS presented low internal consistencies while there was a strong intercorrelation among them. This result suggest both that focusing on just the total score of the scale can be more valid than describing the results according to the scores in each dimension; and that further studies should analyze the factorial solution of the scale in different cultural contexts.” (see limitations and further research, p. 10, lines 385-391).

4.- Thank you for this comment. We are sorry that we did not notice this error, which has been corrected in the new version.